# Tuning singlet oxygen generation with caged organic photosensitizers

Eleni Nestoros[1,2], Fabio de Moliner[1,2], Ferran Nadal-Bufi [1,2], Deborah Seah[1,2], M. Carmen Ortega-Liebana [3,4], Zhiming Cheng[1,2], Sam Benson[1,2], Catherine Adam[3], Larissa Maierhofer [1], Kostiantyn Kozoriz [5], Jun-Seok Lee [5], Asier Unciti-Broceta [3] ✉ & Marc Vendrell [1,2] ✉

Controlling the succession of chemical processes with high specificity in complex systems is advantageous for widespread applications, from biomedical research to drug manufacturing. Despite synthetic advances in bioorthogonal and photochemical methodologies, there is a need for generic chemical approaches that can universally modulate photodynamic reactivity in organic photosensitizers. Herein we present a strategy to fine-tune the production of singlet oxygen in multiple photosensitive scaffolds under the activation of bioresponsive and bioorthogonal stimuli. We demonstrate that the photocatalytic activity of nitrobenzoselenadiazoles can be fully blocked by site-selective incorporation of electron-withdrawing carbamate moieties and restored on demand upon uncaging with a wide range of molecular triggers, including abiotic transition-metal catalysts. We also prove that this strategy can be expanded to most photosensitizers, including diverse structures and spectral properties. Finally, we show that such advanced control of singlet oxygen generation can be broadly applied to the photodynamic ablation of human cells as well as to regulate the release of singlet oxygen in the semisynthesis of natural product drugs.

Organic photosensitizers (PS) are light-absorbing molecules that catalyze and modulate the course of photochemical reactions[1]. The applications of organic PS in chemical sciences are broad and diverse, ranging from solar fuel generation by photocatalytic reduction of water and carbon dioxide[2] to manufacturing of new materials via photopolymerization[3]. One of the most widespread uses of organic PS is the generation of reactive oxygen species (ROS), including singlet oxygen, upon illumination. Singlet oxygen can be also employed as a traceless oxidant for chemoselective reactions[4,5] as well as an inducer of biomolecular oxidation for photodynamic therapy (PDT)[6]. For most applications, non-functionalized chromophores and dyes (e.g., benzoselenadiazoles, phenothiazines) are used as organic PS[7,8], with

restricted control of their photocatalytic activity. Therefore, the design of new strategies to fine tune the ROS and singlet oxygen production by organic PS would open new avenues for their application in chemical and biological processes with enhanced spatiotemporal resolution.

The molecular design of activatable PS allows fine tuning of singlet oxygen production in a chemically dependent manner. Because of the highly conjugated systems featured in most organic chromophores, activatable organic PS can be prepared by altering the electron density within their cores via intramolecular charge transfer (ICT) or photoinduced electron transfer (PeT)[9,10], in a similar fashion to those utilized for bioresponsive fluorophores[11–18]. Some examples of

[1]Centre for Inflammation Research, The University of Edinburgh, Edinburgh, UK. [2]IRR Chemistry Hub, Institute for Regeneration and Repair, The University of Edinburgh, Edinburgh, UK. [3]Edinburgh Cancer Research, Cancer Research UK Scotland Centre, Institute of Genetics and Cancer, University of Edinburgh, Edinburgh, UK. [4]Centre Pfizer-GENYO, Faculty of Pharmacy, University of Granada, Granada, Spain. [5]Department of Pharmacology, College of Medicine, Korea University, Seoul, Korea. ✉e-mail: asier.ub@ed.ac.uk; marc.vendrell@ed.ac.uk

activatable PS include pH-sensitive constructs that switch on upon deprotonation of phenol groups[19,20] or azo-derivatized agents where the reduction of diazo moieties to amines leads to enhanced inter-system crossing and singlet oxygen generation[21–26]. Although other chemical strategies (e.g., supramolecular assembly, aggregation-induced emission, bioorthogonal ligation, AND-gating) have been successfully adapted to the synthesis of switchable PS[27–34], current strategies are limited in terms of stimuli and structural diversity, therefore being incompatible with a broad range of biochemical processes and organic PS.

The incorporation of chemical cages into bioactive molecules has been reported as an effective strategy to modulate their pharmacological activity and minimize off-target effects. For instance, caged prodrugs and profluorophores react with metallic nanomaterials and complexes (e.g., Au, Pd, and Ru, among others) to release active drugs and fluorophores in situ with enhanced therapeutic efficacy and fluorescence brightness, respectively[35–39]. These bioorthogonal strategies can provide a non-biological approach to control when and where xenobiotics are activated; however, they have not yet been generically incorporated into the molecular design of PS due to the lack of universal chemical approaches to fine-tune photodynamic reactivity.

We envisioned that the photocatalytic properties of small molecule PS could be controlled by caging essential electron-rich functional groups that are ubiquitously found in organic PS[40]. In this work, we have designed a generic platform to chemically control the activity of organic PS, spanning diverse chemical structures and optical properties. We observed that the site-selective introduction of electron-withdrawing cages (i.e., carbamates) in amine-functionalized chromophores depletes the electron density within the PS cores and results in substantial changes of their excitation profiles, with large hypsochromic shifts up to 150 nm. Furthermore, we demonstrated that this strategy is adaptable to chemical uncaging under different exogenous and endogenous stimuli, creating new opportunities for bioresponsive and bioorthogonal PDT as well as for the controlled release of singlet oxygen in chemical transformations. The simplicity and versatility of this platform and its compatibility with structurally diverse PS will enable the design of novel responsive systems for controlled generation of singlet oxygen with high spatiotemporal resolution.

## Results

### A caging strategy for amino-containing organic PS

Nitrobenzoselenadiazoles are low molecular weight PS with moderate singlet oxygen generation quantum yields and relevant features for bioprobe design[7,41] (e.g., neutral character, excitation in the visible range, cell permeability). The photosensitive activity of nitrobenzoselenadiazoles relies on the push-pull dipole between its electron-donating amine group at position 4 and the electron-withdrawing nitro group at position 7 (Fig. 1a). Analogously to the use of carbamate cages to mask essential moieties for target binding and pharmacodynamic activity in small molecule drugs[42,43], we envisaged that we could modulate the reactivity of nitrobenzoselenadiazoles by modifying the substitution pattern of amines with caging groups, thus altering ICT and dipolar strength. Furthermore, because electron-donating amino groups are frequent structural features in the heterocyclic cores of numerous PS, this caging approach could be in principle extended to many singlet oxygen generators.

First, we examined the effect of introducing carbamate-based electron-withdrawing groups at position 4 of the nitrobenzoselenadiazole core (Fig. 1a). For these experiments, we employed the tert-butyloxycarbonyl (Boc) group due to its chemical simplicity, accessibility, and stability. We synthesized the nitrobenzoselenadiazole ethylamine compound 1 in high yields (~90%) and batch scales up to 1 g by nucleophilic aromatic substitution of its fluoride precursor using

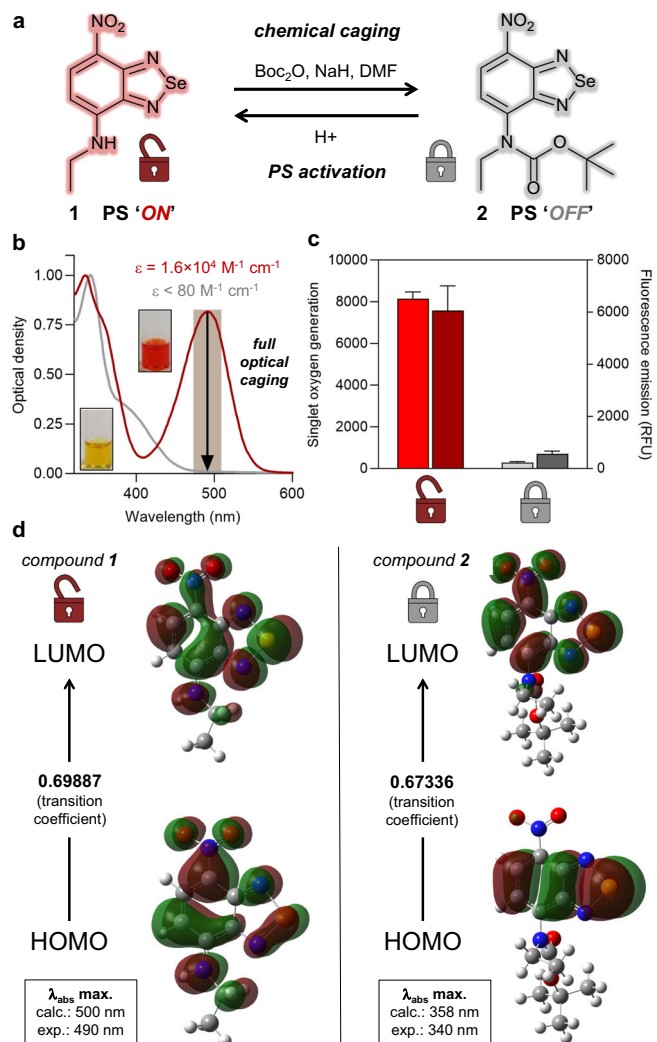

**Fig. 1 | Exploration of carbamate caging for amino-containing PS. a** Chemical structures of uncaged (**1**) and caged (**2**) nitrobenzoselenadiazoles. **b** Normalized absorbance spectra of compounds **1** (dark red) and **2** (gray, both at 200 μM in EtOH) with calculated extinction coefficients at 490 nm. Insets show pictographs of DMSO solutions of compounds **1** and **2** (both at 20 mM, DMSO). **c** Singlet oxygen generation (left) and fluorescence emission (right) of uncaged compound **1** (10 μM, red bars) and caged compound **2** (10 μM, gray bars) measured in EtOH. Compound **1** showed a relative singlet oxygen quantum yield (reference: Rose Bengal) of 55% in EtOH (520 nm, 0.4–0.5 mW cm⁻²). Singlet oxygen generation for compounds **1** and **2** was compared in relative fluorescence units (RFU) by monitoring the fluorescence emission (525 nm) of singlet oxygen sensor green (SOSG, 2.5 μM in EtOH) before and after illumination (520 nm, 10 mW cm⁻²) for 10 min. Values presented as means and error bars as SEM (n = 15). **d** Electron density maps in molecular orbitals for compounds **1** and **2** calculated by DFT. Calculations were performed using the WB97XD functional with the cc-pVTZ basis set and the SMD solvent model in EtOH. The TD-DFT parameters specified the examination of 10 states, focusing on the first excited state. Source data (**c**) are provided as a Source Data file.

ethylamine hydrochloride under basic conditions. The treatment of compound **1** with sodium hydride (NaH) followed by reaction with di-tert-butyl dicarbonate (Boc₂O) rendered the caged compound **2** (Fig. 1a) with overall recovery yields around 40%. Both compounds were isolated by normal phase chromatography with purities over 95% (full synthetic and characterization details in Supplementary Methods).

Next, we compared the optical properties of both selenadiazoles **1** and **2** as representative models of uncaged and caged PS, respectively. The electron-withdrawing effect of the carbamate moiety led to a significant hypsochromic shift in the absorbance profile of the

nitrobenzoselenadiazole core, with maxima wavelengths shifting from 490 nm in compound **1** to 340 nm in compound **2** (Fig. 1b and Supplementary Fig. 1). As a result, whereas the extinction molar coefficient at 490 nm of compound **1** featured standard values for benzodiazoles[41,44,45] (i.e., in the $2 \times 10^4 M^{-1} cm^{-1}$ range), compound **2** displayed complete optical caging and over 2 orders of magnitude lower extinction molar coefficients in the visible range (Fig. 1b, c). This exceptional level of optical quenching via carbamate caging was supported by density functional theory (DFT) calculations. Our findings indicated an increased molecular orbital density at the amine group of compound **1** -in both HOMO and LUMO levels-, which contrasted with the reduced molecular orbital density at position 4 of compound **2** (Fig. 1d). As a result, compound **2** featured a shorter delocalized aromatic system with blue-shifted absorbance maximum wavelengths and a weaker push-pull dipole.

In order to examine whether this optical caging methodology could be utilized to regulate photocatalytic activity, we analyzed the optical properties and production of singlet oxygen of compounds **1** and **2** upon visible light irradiation. While compound **1** showed reasonable fluorescence intensity and singlet oxygen generation, carbamate caging completely abolished both the fluorescence emission and photosensitive activity of the nitrobenzoselenadiazole core (Fig. 1c), highlighting its utility for the molecular design of chemically activatable PS. DFT calculations revealed larger energy gaps for intersystem crossing between the $S^*$ and $T^2$ excited states for the caged compound **2**, which is expected to hinder the transition to the triplet state and subsequent singlet oxygen generation upon light exposure (Supplementary Fig. 2). Importantly, we confirmed the reversibility of this approach by monitoring the uncaging of compound **2** and concomitant photodynamic activation in acidic media. Within 10 min, we observed full conversion of the caged PS **2** to the corresponding compound **1** without formation of side products (Supplementary Figs. 3 and 4) or loss of photocatalytic activity (Supplementary Fig. 5). Altogether, these results indicate that the site-specific introduction of electron-withdrawing carbamate groups is an effective strategy to modulate the photodynamic power of amino-containing nitrobenzoselenadiazoles.

## A chemically diverse collection of bioorthogonal and bioresponsive nitrobenzoselenadiazole PS

We decided to expand our carbamate caging strategy by diversifying the range of activating stimuli, and designed a collection of nitrobenzoselenadiazole PS that could be uncaged by both exogenous and endogenous molecular triggers. For the former, we incorporated propargyloxycarbonyl (Poc) and allyloxycarbonyl (Alloc) moieties (compounds **3** and **4** respectively, Fig. 2) that bioorthogonally react with various abiotic transition metals[39,46–49], as well as 6-nitroveratryloxycarbonyl (Nvoc) groups that can be photocleaved under UV irradiation (compound **5**, Fig. 2)[50]. To create caged PS that are responsive to various biostimuli, we designed the bioresponsive PS **6** and **7** (Fig. 2) to be activated in response to oxidative stress (i.e., via reaction with $H_2O_2$ or $H_2S$) as well as compound **8** for uncaging in hypoxic conditions that accelerate the reduction of azo bonds[51]. Altogether, this library was designed to explore the true versatility of our chemical caging strategy and its compatibility with different physical, bioorthogonal and biochemical triggers for a diverse set of applications.

The caged bioorthogonal PS **3, 4** and **5** were synthesized by straightforward reaction of the nitrobenzoselenadiazole **1** with the corresponding commercial chloroformates in good yields around 50% (full synthetic and characterization details in Supplementary Methods). Given the limited availability of chloroformates for the preparation of the bioresponsive PS **6–8**, we optimized a synthetic procedure to form carbamate groups by reacting the amino-containing nitrobenzoselenadiazole **1** with the corresponding activated carbonates. First, we isolated the 4-nitrophenylcarbonates **9, 10**

and **11** by coupling 4-nitrophenylchloroformate with readily accessible benzyl alcohols (i.e., 4-(hydroxymethyl)phenylboronic acid pinacol ester for carbonate **9**, 4-azidobenzyl alcohol for carbonate **10** and 4-phenylazobenzyl alcohol for carbonate **11**, Fig. 2). Next, we reacted all three carbonates with nitrobenzoselenadiazole **1** under mild microwave irradiation to obtain the corresponding caged PS (Fig. 2). Importantly, all PS (**3–8**) were isolated by normal phase chromatography with overall yields between 16 and 55% and high purities over 95% (full synthetic and characterization details in Supplementary Methods), highlighting the modularity of this synthetic strategy.

Having synthesized the collection of caged PS, we next examined their uncaging mechanism by HPLC-MS (Supplementary Figs. 6–11) and fluorescence analysis ($\lambda_{exc/em}$: 490/620 nm). Upon reaction with selected abiotic transition metals (e.g., Pd and Au-functionalized resins as examples of biocompatible microimplants[35,52], Pd(0)-complex stabilized with triphenylphosphine ligands[53]), caged PS **3** and **4** were successfully converted to the active PS **1** via metal-triggered dealkylation followed by $CO_2$ release; similarly, the photocleavable PS **5** was uncaged under blue light (365 nm) irradiation (Fig. 3a and Supplementary Fig. 12). Furthermore, we corroborated the reactivity and turn-on effect of the bioresponsive PS **6–8** upon incubation with increasing concentrations of oxidants and reducing agents (i.e., $H_2O_2$ for the boronate caged compound **6**, $H_2S$ for the azido caged compound **7** and $N_2H_4$ for the azo caged compound **8**) (Fig. 3a). In this set of compounds, the oxidation (of a boronic acid group to a phenol group for compound **6**) or reduction (of azide or azo groups to amines in compounds **7** and **8**, respectively) was followed by 1,6-elimination and $CO_2$ release to form the active nitrobenzoselenadiazole moiety (Fig. 3a).

To study the functionality of our strategy, we measured the singlet oxygen production of all caged PS before and after incubation with their respective triggers. For these experiments, we monitored the absorbance readout at 410 nm of the singlet oxygen trap 1,3-diphenylisobenzofuran (DPBF) in the absence and presence of trigger and before and after 520 nm LED illumination (Fig. 3c). As expected, none of the caged PS **3–8** produced detectable levels of the active PS **1** and concomitant singlet oxygen generation after 520 nm LED irradiation; however, all compounds were decaged after incubation with their respective triggers and generated high levels of singlet oxygen upon photostimulation. Notably, for most compounds, the percentages of released singlet oxygen upon activation were comparable to those produced by the nitrobenzoselenzadiazole **1**, proving the reversibility of the carbamate caging strategy and applicability to a broad range of chemical and physical stimuli. Furthermore, we confirmed that all uncaging reactions were compatible with aqueous media and physiological conditions by in vitro assays with biologically-relevant analytes (Supplementary Fig. 13) and by confocal microscopy experiments in human MCF-7 cells, where intracellular fluorescence signals resulting from the uncaging of the nitrobenzoselenadiazoles **3-8** were only observed upon reaction with their respective triggers (Supplementary Fig. 14).

## Bioorthogonal control of the cellular permeability of nitrobenzoselenadiazole PS and downstream PDT in cancer cells

We next decided to explore whether the carbamate caging strategy could be used not only to control singlet oxygen generation but also to modulate the physicochemical properties and distribution of organic PS. Numerous reports have described that the generation of singlet oxygen within intracellular environments is more effective at inducing apoptosis and cancer cell death[54]; therefore, we rationally designed a cell impermeable caged nitrobenzoselenadiazole PS **12** (Fig. 4a) that could enter and kill cancer cells only after reaction with a bioorthogonal trigger. Compound **12** included the modification of the nitrobenzoselenadiazole core with a benzyloxycarbonyl group featuring a negatively charged sulfonate group that prevented cell entry and a metal-responsive propargyloxy moiety (Fig. 4a and Supplementary

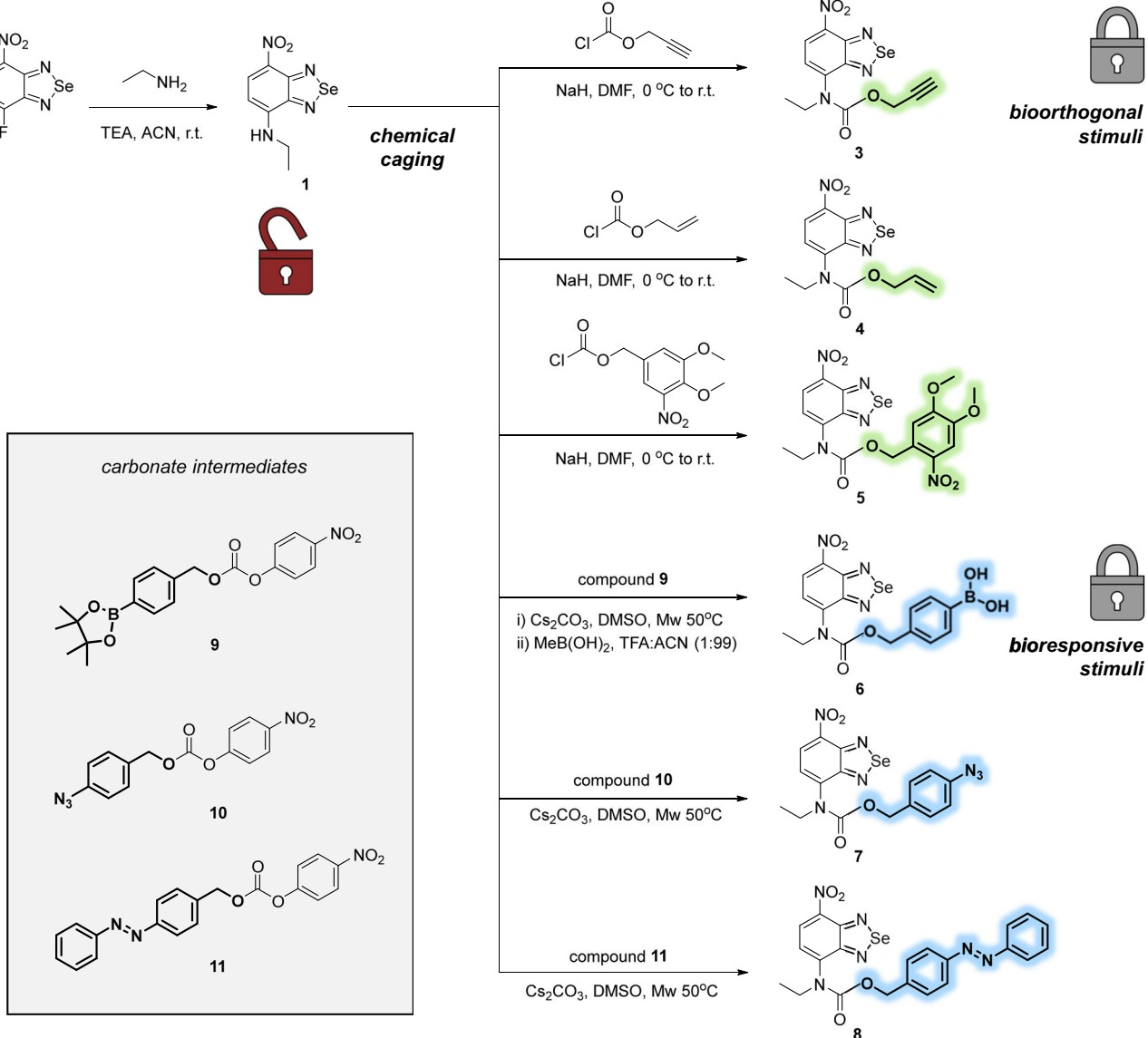

**Fig. 2 | Synthesis of a library of bioorthogonal and bioresponsive nitrobenzoselenadiazole PS.** Chemical structures and synthetic procedures for the preparation of bioorthogonal caged PS (compounds **3–5**, caging groups highlighted in green) and bioresponsive caged PS (compounds **6–8**, caging groups highlighted in blue). Inset: chemical structures of 4-nitrophenylcarbonates (**9–11**) as precursors for the preparation of the caged nitrobenzoselenadiazoles (**6–8**).

Fig. 15). Using this design, we envisioned that the metal-mediated O-depropargylation would result in a 1,4-elimination with subsequent $CO_2$ release and in situ formation of the cell permeable and photoactive nitrobenzoselenadiazole **1** (Fig. 4a).

We synthesized compound **12** in two steps. First, we performed the reaction between the amine-containing nitrobenzoselenadiazole **1** and a 4-nitrophenylcarbonate featuring two propargyloxy groups under basic conditions and microwave irradiation to afford the intermediate carbamate **13** (Supplementary Fig. 15). Second, compound **13** was subjected to copper-catalyzed cycloaddition with the negatively charged 2-azidoethanesulfonate to render the hydrophilic and cell impermeable compound **12**. Of note, despite the presence of two chemically equivalent alkyne moieties in compound **13**, the formation of the double triazole adduct was minimized by controlling the stoichiometry (alkyne:azide, 1.5:1), the reaction time (20 min) and the choice of ligand (L-histidine). Compound **12** was isolated in purities >95% after HPLC purification (full synthetic and characterization details in Supplementary Methods).

Next, we confirmed the excellent water solubility of compound **12** (Supplementary Fig. 16) and its effective optical quenching, as displayed by the characteristic large hypsochromic shift in absorbance maximum of around 150 nm (Supplementary Fig. 17). Encouraged by the enhanced catalytic activity and biocompatibility of AuPd nanoalloys[55], we also prepared a new class of Tentagel-based AuPd resins to perform extracellular PS uncaging studies (see preparation details and characterization in Supplementary Methods and Supplementary Fig. 18). Notably, compound **12** was fully cleaved in aqueous media after overnight incubation with AuPd resins at 37 °C to release the photoactive nitrobenzoselenadiazole PS **1**, as confirmed by HPLC-MS analysis (Fig. 4b). Furthermore, we performed confocal microscopy experiments in MCF-7 cancer cells to confirm that uncaging of compound **12** with AuPd resins released the cell-permeable compound **1** (Supplementary Fig. 19), thus validating the utility of this caging strategy to modulate the physicochemical properties of nitrobenzoselenadiazole PS. Finally, we examined whether the bioorthogonal activation of compound **12** could be applied to tune PDT-mediated

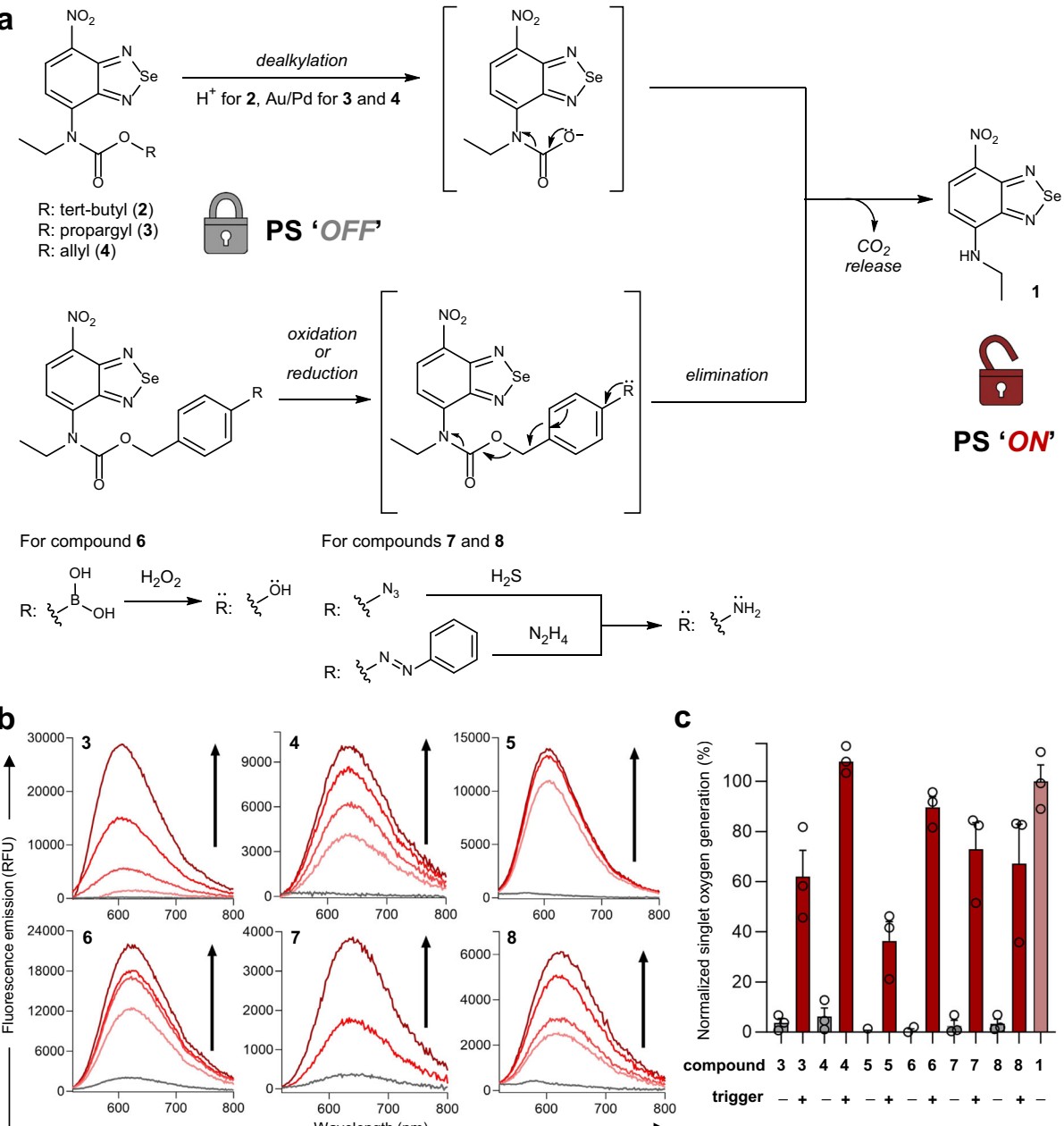

**Fig. 3 | Mechanism of action and in vitro characterization of bioorthogonal and bioresponsive nitrobenzoselenadiazole PS. a** Uncaging mechanisms of caged PS involve dealkylation (top) or oxidation/reduction followed by elimination (bottom) to release the photoactive nitrobenzoselenadiazole scaffold. **b** Fluorescence analysis ($\lambda_{exc/em}$: 490/620 nm) for compounds **3**–**8** (25 μM, pH 7.4) alone (gray lines) or under different triggers (red lines; for **3**, Au resins up to 10 mg mL$^{-1}$ (15 h); for **4**, Pd(PPh$_3$)$_4$ up to 25 μM (15 h); for **5**, illumination 365 nm at 1 mW cm$^{-2}$ up to 20 min; for **6**, H$_2$O$_2$ at 25-200 μM (15 h); for **7**, H$_2$S up to 5 mM (15 h); for **8**, N$_2$H$_4$, up to 100 μM (15 h); see details in Methods). **c** Singlet oxygen generation for all caged PS (50 μM) (normalized against compound **1**) measured as percentages of absorbance decrease of DPBF (300 μM in EtOH) before reaction (gray bars) and after reaction (red bars) with the trigger followed by illumination at 520 nm (0.5 mW cm$^{-2}$, 6−10 min). Values presented as means and error bars as SEM ($n = 3$, independent experiments). Source data (**c**) are provided as a Source Data file.

cancer cell ablation in multiple cell lines of diverse origin, namely MCF-7 (estrogen receptor-positive breast cancer), MDA-MB-231 (triple negative breast cancer) and Ln18 (glioblastoma)[56–58]. All cell lines were incubated under physiological conditions with compound **12** and AuPd resins, followed by light illumination to trigger the production of intracellular singlet oxygen. As shown in Fig. 4c, only the combination treatment (i.e., compound **12**, AuPd resins and light) resulted in substantial cell death, with toxicity percentages averaging 70% in the different cancer types. Under these conditions, we observed notable ROS production (Supplementary Fig. 20), which led to cellular apoptosis as confirmed by positive AnnexinV (Supplementary Fig. 21),

caspase-3 staining (Supplementary Fig. 22) and flow cytometry analysis (Supplementary Fig. 23). Notably, the individual treatment of cancer cells with compound **12** and AuPd resins or light irradiation (but not both) had minimal impact on cell viability, highlighting the utility of bioorthogonal caging to fine-tune the PDT activity of organic PS.

## A universal approach for caging UV-to-NIR photosensitive structures
In order to assess the modularity and versatility of our caging strategy, we expanded it to produce a palette of chemically diverse activatable PS with different excitation wavelengths. Specifically, we designed

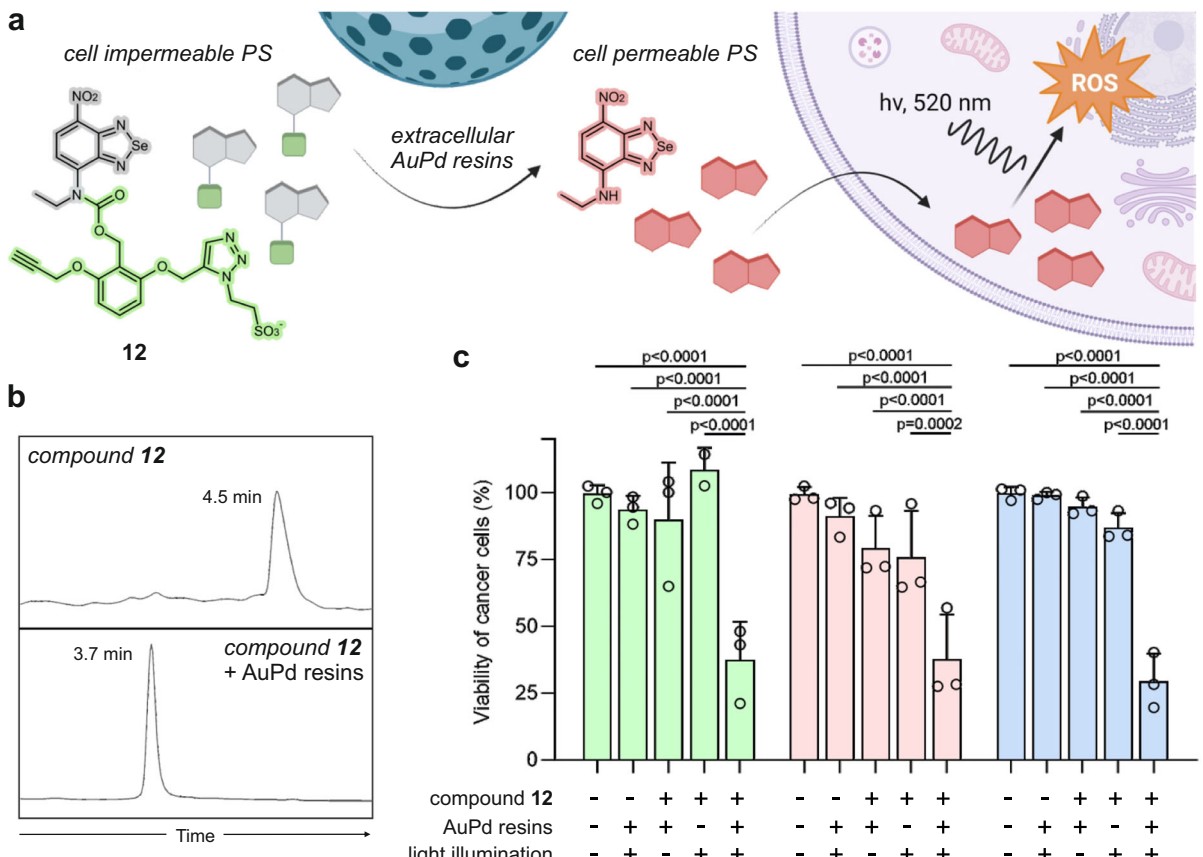

**Fig. 4 | Cancer cell ablation by fine-tuning the cell permeability of nitrobenzoselenadiazole PS. a** Schematic illustration of compound **12** and mechanism for bioorthogonal activation of PDT. The cell impermeable PS is uncaged by extracellular AuPd resins and releases the active PS, which enters cancer cells and generates ROS upon light exposure, leading to cell death. **b** Representative HPLC-MS traces of compound **12** (300 μM) before and after reaction with AuPd resins (5 mg mL$^{-1}$) in PBS for 16 h. Experimental m/z values: 688.0 (caged compound **12** [M+Na]$^+$), 273.0 (uncaged compound **1** [M + H]$^+$). **c** Cell viabilities of MCF-7 (green), MDA-MB-231 (red) and Ln18 (blue) cancer cells after

treatment with 400 μM of compound **12**, 1 mg mL$^{-1}$ AuPd resins, and illumination with white light (10 mW cm$^{-2}$) for 60 min under a UV filter. Cell viabilities were determined by the resazurin assay using PBS and 0.01% (v/v) Triton X−100 as negative and positive controls, respectively. Values presented as means and error bars as SD ($n = 3$, independent experiments). $P$ values were calculated using a two-way ANOVA. Source data (**c**) are provided as a Source Data file. **a** created with BioRender.com, released under a Creative Commons Attribution-NonCommercial-NoDerivs 4.0 International license.

a collection of caged PS based on 2-thioxocoumarins[59] (**14**), thionaphthalimides[60] (**15**), benzophenoxazines[61] (**16**) and phenothiazines[62] (**17**), which display absorbance wavelengths across the entire range of the visible spectrum (Fig. 5a). 2-Thioxocoumarins were prepared from the commercially available 7-amino-4-methylcoumarin via a two-step synthesis (Supplementary Fig. 24). After the formation of carbamates using conventional procedures[63,64] (e.g., Boc for compound **14a**, Poc for compound **14b**), the carbonyl group of the coumarin scaffold was converted into a thiocarbonyl in a straightforward replacement promoted by Lawesson's reagent[65]. On the other hand, caged thionaphthalimides were obtained through a divergent strategy starting from 4-bromo-1,8-naphthalic anhydride. First, the anhydride was converted to the corresponding aminopropyl-substituted naphthalimide and refluxed in toluene in the presence of Lawesson's reagent to afford the thionaphthalimide core in quantitative yields (Supplementary Fig. 24). Second, the thionaphthalimide was caged by reaction with NaH and addition of di-tert-butylcarbonate or propargyl chloroformate (e.g., Boc for compound **15a**, Poc for compound **15b**). Finally, benzophenoxazines (e.g., Boc for compound **16a**, Poc for compound **16b**) and phenothiazines (e.g., Boc for compound **17a**, Poc for compound **17b**) were synthesized in one-step reactions from their commercially available precursors Nile Blue (for compounds **16a** and **16b**) and azure B (for compounds **17a** and **17b**) and the corresponding di-tert-butylcarbonate or propargyl chloroformate

(Supplementary Fig. 24). All caged PS were isolated in purities over 95% and widely ranging yields (e.g., from 5% for compound **16a** to quantitative yields for compound **17a**; full synthetic and characterization details in Supplementary Methods).

Having synthesized a collection of caged organic PS, we measured their absorbance spectra and compared them to the corresponding uncaged PS. All caged PS (compounds **14a/b**, **15a/b**, **16a/b** and **17a/b**) displayed substantially shorter absorbance maxima wavelengths than their uncaged counterparts (Fig. 5b), with hypsochromic shifts ranging from 25 nm (i.e., 425 nm to 400 nm) for 2-thioxocoumarin **14** to 140 nm (i.e., 580 nm to 440 nm) for thionaphthalimide **15** (Fig. 5b and Supplementary Fig. 25). Next, we analyzed the bioorthogonal uncaging of all Poc-protected PS (**14b-17b**) using AuPd resins. As expected, we observed rapid and clean deprotections for the benzophenoxazine **16b** (Supplementary Fig. 26) and the phenothiazine **17b** (Supplementary Fig. 27). Interestingly, no uncaging was detected for the two thiocarbonyl-containing PS (i.e., the 2-thioxocoumarin **14b** and the thionaphthalimide **15b**), likely due to the potential poisoning of AuPd resins by complexation with thiocarbonyls[39,55,66]. To investigate this point, we further analyzed the reactivity of caged 2-thioxocoumarins (i.e., compounds **14a** and **14b**) in detail (Supplementary Note 1). Using acidic conditions as the chemical trigger, the Boc-caged thionaphthalimide **15a** was readily uncaged to release the corresponding PS **15** (Supplementary Fig. 28). Finally, we measured the singlet oxygen

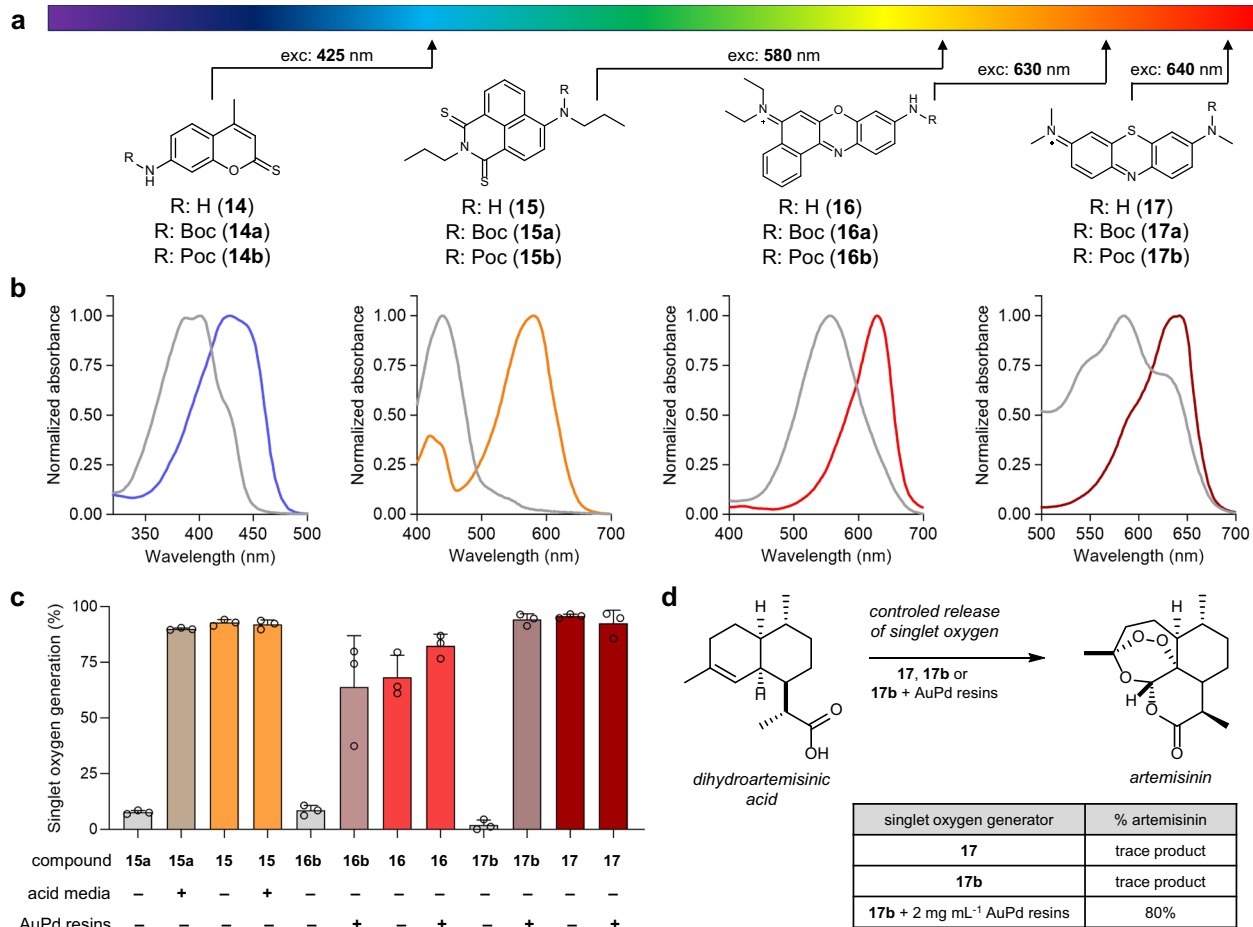

**Fig. 5 | A general strategy for caging UV-to-NIR PS. a** Structures of chemically diverse organic PS (with their corresponding excitation maxima) and the synthesized caged analogs. **14**: 2-thioxocoumarin, **15**: thionaphthalimide, **16**: benzophenoxazine, **17**: phenothiazine. **b** Normalized absorbance spectra of caged compounds (gray lines) and their uncaged PS (from left to right, **14**: blue, **15**: orange, **16**: red, **17**: dark red; all at 100 μM in EtOH). **c** Singlet oxygen generation for compounds **15-17** and their caged analogs measured by monitoring the percentage of absorbance decrease of DPBF (300 μM in EtOH) before (gray bars) and after (colored bars) incubation with acid media (TFA:DCM, 1:1) or 1-10 mg mL$^{-1}$ AuPd resins followed by illumination (**15** (orange): 580 nm, 0.4 mW cm$^{-2}$, 1 min; **16** (red):

640 nm, 1.5 mW cm$^{-2}$, 40 min; **17** (dark red): 640 nm, 0.4 mW cm$^{-2}$, 3 min). Values presented as means and error bars as SD ($n = 3$, independent experiments). **d** Singlet oxygen-mediated transformation of dihydroartemisinic acid into the antimalarial drug artemisinin. Conversion rates by HPLC-MS measurements under the same experimental conditions (640 nm illumination, all PS at 10% mol) using uncaged PS **17** (1 mM) or the caged PS **17b** (1 mM) without or with AuPd resins (2 mg mL$^{-1}$). Time-course analysis of the chemical transformation using caged PS **17b** and AuPd resins is shown in Supplementary Fig. 31. Source data (**c**) are provided as a Source Data file.

production of these caged PS (**15a, 16b** and **17b**) before and after incubation with the respective triggers and illumination at the appropriate wavelengths. As expected, none of the caged PS produced substantial levels of singlet oxygen while all compounds restored the singlet oxygen generation levels of the corresponding uncaged PS (i.e., **15**, **16** and **17**) after uncaging (Fig. 5c). These results corroborate that our caging strategy can be adapted to a wide range of amine-containing organic PS with variable chemical structures and optical properties.

Finally, we decided to examine whether compound **17b** could be used to improve the outcomes of chemical transformations involving singlet oxygen. Phenothiazine PS have been widely reported in the literature as a reagent for light-induced chemical oxidations[67]; however, their utility is limited when chemical reactions require controlled release of singlet oxygen[68]. One prime example of a complex reaction cascade promoted by singlet oxygen is the semi-synthesis of the anti-malarial drug artemisinin from its precursor dihydroartemisinic acid (Fig. 5d). Although a route for the preparation of artemisinin from di-hydroartemisinic acid was reported as early as 1989[69], this transformation is challenging due to poor uniformity of the singlet oxygen

production during the reaction time. Some elegant flow chemistry approaches have been reported to address this challenge, yet they require specialized equipment[70]. We envisioned that the controlled release of the PS **17** by AuPd resin-mediated uncaging of its protected analog **17b** would represent a simple alternative to continuously feeding the reaction vessel with fresh compound **17** for the chemical synthesis of artemisinin in a one-pot manner. After optimization of the reaction conditions (Supplementary Note 2 and Supplementary Fig. 29), we found that irradiation (640 nm, 0.8 mW cm$^{-2}$) of a water:methanol solution of dihydroartemisinic acid in the presence of compound **17b** (10% mol) and AuPd resins (2 mg mL$^{-1}$) afforded very high conversion rates to artemisinin (~80% after 10 h; Fig. 5d and Supplementary Fig. 30). Importantly, we also observed gradual uncaging of compound **17b** in the reaction mixture, supporting the advantageous effect of controlled singlet oxygen generation (Supplementary Fig. 31). As controls, we performed analogous reactions under the same illumination regime using a single addition of equimolar amounts of uncaged PS **17** or compound **17b** in the absence of AuPd resins and found only traces of artemisinin in the reaction mixtures (Fig. 5d). Finally, to mimic the continuous generation of PS **17** in the uncaging approach, the reaction

was carried out under irradiation by adding the phenothiazine **17** in a portion-wise manner (from 0.1 mM to 1 mM, 10% increment every hour). Under these conditions, the reaction yields improved to 69% (Supplementary Fig. 32), in agreement with the hypothesis that singlet oxygen production is enhanced by adding or in situ generating fresh compound **17** during the course of the reaction. These results confirmed the importance of controlling singlet oxygen generation in some chemical transformations and indicate that our PS caging strategy can be employed to modulate complex photocatalyzed chemical reactions that require fine tuning of singlet oxygen production. The in situ and gradual production of sensitive reagents within a closed reactor could be particularly helpful for procedures where the portion-wise addition of the reagent is difficult, such as reactions under inert atmosphere.

## Discussion

Cells control the reactivity of singlet oxygen by orchestrating the action of different biomolecules during photosynthesis or in the natural production of ROS by redox processes, generating useful products that feed into its complex network of biochemical events. Researchers have learned to harvest the unique reactivity of singlet oxygen into multiple photochemical oxidative processes, thus becoming an integral reactant for the synthesis of materials and small organic molecules, and for biomedical uses such as PDT. Due to its high reactivity, however, it would be beneficial to attain further spatiotemporal control over the production of singlet oxygen by photoactivated PS. For example, to reduce skin damage by visible-light excitable PS or to sustain the generation of the oxidative species by the time-controlled release of a PS. Herein we report a generic platform for the chemical tuning of singlet oxygen generation using caged organic PS that respond to a broad array of bioorthogonal and bioresponsive stimuli. We demonstrated that the masking of electron-donating amines in structurally diverse PS switches off their photoexcitability and thereby the generation of ROS upon light irradiation. Importantly, the photocatalytic activity was restored upon selective uncaging of the PS with different triggers. The broad applicability of this strategy was proven with a library of nitrobenzoselenadiazoles including a range of carbamate functionalities responding to different chemical and biological stimuli. Furthermore, we used solid-supported metal catalysts, which are compatible with standard synthesis reactors, microfluidic systems, cell culture and living organisms, to modulate the cell entry and downstream PDT activity of caged PS in cancer cells. Finally, we explored the applicability of this strategy for manufacturing processes and showed that the controlled release of phenothiazines can be leveraged to improve the conversion of photocatalyzed chemical reactions, such as the production of the antimalarial drug artemisinin from its precursor dihydroartemisinic acid. Altogether, this generic caging strategy represents a novel approach to fine-tune the production of singlet oxygen in different organic photosensitizers and with multiple biocompatible stimuli, enabling on-demand control of photocatalytic activity in biological systems and synthetic chemical transformations.

## Methods

### ROS measurements and singlet oxygen generation

ROS measurements were performed in solvent-resistant 96-well plates. 1,3-diphenylisobenzofuran (DPBF) was added at the indicated concentrations and the absorbance (410 nm) was measured before and after illumination (520 nm, 0.45–0.5 mW, 6–10 min). The relative percentages of decrease in absorbance were calculated using the following formula:

$$\text{Abs decrease}(\%) = [(\text{Abs}_{Sb} - \text{Abs}_{Sa})(\text{Abs}_{Sb})^{-1}$$
$$- (\text{Abs}_{Bb} - \text{Abs}_{Ba})(\text{Abs}_{Bb})^{-1}] \times 100$$

where $\text{Abs}_{Sb}$ and $\text{Abs}_{Sa}$ are the respective absorbance values at 410 nm for the sample before and after illumination, and $\text{Abs}_{Bb}$ and $\text{Abs}_{Ba}$ are the respective absorbance values at 410 nm for the appropriate blank before and after illumination. Alternatively, singlet oxygen production was measured using Singlet Oxygen Sensor Green (SOSG) in EtOH at the indicated concentrations. Fluorescence intensity (exc/em = 488/525 nm) was measured before and after illumination (520 nm, 10 mW) for 10 min. Controls containing PS and SOSG alone were also measured.

### Fluorescence emission assays

The fluorescence emission (exc/em = 490/620 nm) for all nitrobenzoselenadiazole compounds (25 µM) were measured under different trigger conditions. Specifically, HCl 4 N (compound **2**); Au resins 0-10 mg mL$^{-1}$ in PBS:MeOH (1:1) (compound **3**); Pd(PPh$_3$)$_4$ 0-25 µM in PBS (compound **4**); 365 nm (1 mW) for 0–30 min in DMSO (compound **5**); H$_2$O$_2$ 0–200 µM (compound **6**); H$_2$S 0–5 mM in PBS:MeOH (1:1) (compound **7**); N$_2$H$_4$ 0–100 µM in PBS:MeOH (1:1) (compound **8**).

### HPLC-MS analysis

HPLC-MS analysis was performed in an Agilent Technologies HPLC system consisting of a 1220 Infinity II autosampler, a diode array detector and a MS detector with an electrospray ionization source and nitrogen as the nebulizer gas. Briefly, aliquots of the reaction mixtures were diluted in DMSO:ACN (1:1) and filtered (PTFE filter, 0.2 mm) before injection. Samples were run using H$_2$O (0.1% HCOOH) and ACN (0.1% HCOOH) as eluents in a gradient method [typically 5% to 95% ACN (0.1% HCOOH) over 8 min] and a flow rate of 1.5 mL min$^{-1}$. Analytical column: Kinetex C$_{18}$ 50 × 4.6 mm$^2$.

### Cell imaging

MCF-7 cells were resuspended in growth medium containing 10% (v/v) FBS, counted using a Countess II FL, plated (10,000 cells well$^{-1}$) in a µ-Slide 18-well glass bottom chamber (IBIDI®), and incubated at 37 °C with 5% CO$_2$ for 24 h. Afterwards, the medium was removed by suction and replaced with 100 µL of phenol red-free medium containing all compounds (50 µM) that had been incubated in PBS in presence of their respective triggers at 37 °C with 5% CO$_2$ for 12 h. Cells were imaged in a Leica SP8 fluorescence confocal microscope (exc/em: 488/620 nm) equipped with a live-cell imaging stage. Fluorescence and brightfield images were acquired using a HC PL APO CS2 20x/0.75 dry lens. Images were acquired and processed with the corresponding microscope software, Leica Application Suite X (LAS X) V1.4.6. For compound **12**, MCF-7 cells were incubated or not with AuPd resins (1 mg mL$^{-1}$) and compound **12** (400 µM) for 24 h, and washed three times with phenol red-free medium before illumination or not with visible light (10 mW cm$^{-2}$, 60 min). For imaging of cell death, cells were incubated with Annexin V-AF647 (1 µg mL$^{-1}$) for 30 min and imaged in a Leica SP8 fluorescence confocal microscope (exc/em: 650/665 nm for Annexin V-AF647). Fluorescence and brightfield images were acquired using a HC PL APO CS2 40x/1.30 oil lens. For imaging of ROS production, cells were incubated with CellROX™ Deep Red (5 µM) for 30 min, washed, and imaged in a Leica SP8 fluorescence confocal microscope (exc/em: 644/665 nm). Fluorescence and brightfield images were acquired using a HC PL APO CS2 20x/0.75 dry lens. For imaging of caspase activity, cells were incubated with CellEvent™ Caspase-3/7 (5 µM) for 30 min, washed, and imaged in a Leica SP8 fluorescence confocal microscope (exc/em: 502/530 nm). Fluorescence and brightfield images were acquired using a HC PL APO CS2 20x/0.75 dry lens.

### Cell viability assays

On day 1, cells were re-suspended in growth medium (RPMI for MCF-7; DMEM for MDA-MB-231 and Ln18) containing 10% (v/v) FBS, counted using a Countess II FL, plated (5000 cells well$^{-1}$) in a Nunc Edge 96-well

plates, and incubated at 37 °C with 5% $CO_2$ for 24 h. On day 2, medium was removed by suction and replaced with 100 μL of fresh medium containing or not AuPd resins (1 mg mL$^{-1}$) and compound **12** (400 μM). On day 3, cells were washed three times with fresh medium and illuminated or not with visible light (10 mW cm$^{-2}$, 60 min). On day 4, cell viability was measured using a resazurin-based assay following the manufacturer's instructions. PBS and 0.01% (v/v) Triton X-100 were used as negative and positive controls, respectively.

### Flow cytometry assays

On day 1, MCF-7 cells were re-suspended in RPMI including FBS (10%, v/v), counted using a Countess II FL, plated (100,000 cells well$^{-1}$) in Nunc Edge 12-well plates, and incubated at 37 °C with 5% $CO_2$ for 24 h. On day 2, medium was removed by suction and replaced with 100 μL of fresh medium containing or not AuPd resins (1 mg mL$^{-1}$) and compound **12** (400 μM). On day 3, cells were washed three times with fresh medium and illuminated or not with visible light (10 mW cm$^{-2}$, 60 min). On day 4, cells were gently scrapped, washed with HEPES-NaCl containing 0.1% BSA and 2 mM $CaCl_2$, and stained with Annexin V-Pacific Blue (1 μg mL$^{-1}$) for 30 min at r.t. Staining with DRAQ7 (3 μM) was performed directly before analysis. Fluorescence emission was acquired on the NovoCyte® under the following excitation/emission filters: Pacific Blue-Annexin V (405/450 nm), DRAQ7 (644/780 nm). Data was analyzed using the FlowJo X software.

### Reporting summary

Further information on research design is available in the Nature Portfolio Reporting Summary linked to this article.

## Data availability

The data generated in this study are provided in the Supplementary Information and in the Source Data file. Any additional data from Figs. 1–5 and Supplementary Figs. 1–61 are available from the corresponding authors upon request. Source data are provided with this paper.

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

## Acknowledgements

This work was supported by the European Commission (H2020-MSCA–IF–2018-841990, M.C.O.L.), the Engineering and Physical Sciences Research Council (EP/W015706/1, M.V.; EP/S010289/1, A.U.B.; EP/N021134/1, A.U.B.), the Medical Research Council (MR/R01566X/1, M.V.),

and an ERC Consolidator Grant (DYNAFLUORS, 771443, M.V.). (S)TEM characterization (FEI-Titan) was supported by the H2020 program (ESTEEM3, 823717, A.U.B.).

## Author contributions

E.N., F.M., and L.M. conducted chemical synthesis and characterization; E.N., F.N.B., D.S., Z.C., and S.B. performed spectroscopic and biological experiments including data analysis; M.C.O.L. and C.A. provided materials for bioorthogonal activation; K.K. and J.S.L. performed computational analysis experiments; A.U.B. and M.V. conceived and designed the study, and wrote the manuscript with feedback from all authors.

## Competing interests

The authors declare no competing interests.
