## [Peer Review File · Nature Communications]

Tuning singlet oxygen generation with caged organic photosensitizersEditorial note: This manuscript has been previously reviewed at another journal that is not operating a transparent peer review scheme. This document only contains reviewer comments and rebuttal letters for versions considered at *Nature Communications*. Mentions of the other journal have been redacted.

REVIEWERS' COMMENTS

Reviewer #1 (Remarks to the Author):

In the revised version, the authors have addressed my previous concerns, leading to a substantially improved paper. Their revisions have not only enhanced the manuscript's overall quality but also significantly clarified its findings. Therefore, I am pleased to recommend this manuscript for publication.

Reviewer #2 (Remarks to the Author):

In this revision, Nestoros et al. replied to my concerns, many of which were shared by other reviewers, in particular regarding novelty. Since the manuscript is now being considered for the journal that I suggested in my first evaluation, I support publishing it. The authors have also added some data and explanations that improve the quality of the paper.

My only remaining concern is the claim that photoactivation of compound 17b is superior to using only compound 17 in the oxidation of dihydroarteminic acid. Since photoactivation of compound 17b gives compound 17, there is no chemical reason why 17 alone would not have similar results provided that the reaction conditions are optimized. The authors themselves now show that this is true in Figures S31 and S32, which clearly show a similar trend for both reactions. Whereas the yield (of a single experiment? –otherwise, please provide statistics) is slightly higher with 17b (82% vs 69% with 17, both at 10 h), it is likely that the yield of the reaction using 17 could be higher with optimized reaction conditions. In my opinion, the results of this experiment do not support the hypothesis of the authors, and I recommend removing it from the manuscript or at least moving it to the supporting information. Otherwise, I fear that this experiment is just distracting and weakens an

otherwise interesting and well-presented paper.

Reviewer #3 (Remarks to the Author):

The comments from this reviewer have been adequately addressed.

Reviewer #4 (Remarks to the Author):

I appreciate the authors' additional work to demonstrate the stability of the probes and their explanations on the versatility of their system. I also agree that the scope of the strategy may be broader than it initially appears. However, I remain unconvinced that the manipulation of fluorescence response and photoinduced singlet oxygen generation of organic dyes through amine-caging possesses the novelty and innovative elements required for publication in [journal name redacted] or Nature Communications. This is primarily because the underlying chemistry has been well established. While the work was competently performed and the results clearly presented, I believe that the manuscript would be better suited for a more specialized journal focusing on chemical biology or organic chemistry.

Response to Reviewers

Reviewer 1

In the revised version, the authors have addressed my previous concerns, leading to a substantially improved paper. Their revisions have not only enhanced the manuscript's overall quality but also significantly clarified its findings. Therefore, I am pleased to recommend this manuscript for publication.

Answer: We thank the reviewer for the constructive criticisms and positive comments about our work.

Reviewer 2

In this revision, Nestoros et al. replied to my concerns, many of which were shared by other reviewers, in particular regarding novelty. Since the manuscript is now being considered for the journal that I suggested in my first evaluation, I support publishing it. The authors have also added some data and explanations that improve the quality of the paper.

My only remaining concern is the claim that photoactivation of compound 17b is superior to using only compound 17 in the oxidation of dihydroarteminic acid. Since photoactivation of compound 17b gives compound 17, there is no chemical reason why 17 alone would not have similar results provided that the reaction conditions are optimized. The authors themselves now show that this is true in Figures S31 and S32, which clearly show a similar trend for both reactions. Whereas the yield (of a single experiment? –otherwise, please provide statistics) is slightly higher with 17b (82% vs 69% with 17, both at 10 h), it is likely that the yield of the reaction using 17 could be higher with optimized reaction conditions. In my opinion, the results of this experiment do not support the hypothesis of the authors, and I recommend removing it from the manuscript or at least moving it to the supporting information. Otherwise, I fear that this experiment is just distracting and weakens an otherwise interesting and well-presented paper.

Answer: We are grateful to the Reviewer for the positive comments. Regarding the synthesis of artemisinin, we believe that -despite being one single example- the results are very relevant to this work. Our findings show that the photosensitizer **17** was unable to oxidize dihydroarteminic acid into artemisinin in a closed reaction vessel under UV irradiation (only traces of product were produced). In contrast, the *in situ* gradual transformation of optically-inactive **17b** into **17** by AuPd-resins, with concomitant light irradiation, led to good reaction yields (>80%). In fact, in the last revision we added new data demonstrating that, to achieve similar yields to those of our strategy but with the direct use of photosensitizer **17**, this reagent needs to be added portion-wise along the duration of the reaction (10 % increments per hour). We followed this procedure to mimic the conditions of a continuous flow chemistry reactor. Altogether, our results provide significant evidence that fine-tuning of singlet oxygen production is essential in this synthetic process. We reckon that strategies capable of generating a fresh reagent 'in situ' in a continuous manner would be particularly useful for procedures where portion-wise addition of the reagent is difficult, such as air-sensitive or moisture-sensitive reactions, and therefore this approach has remarkable value for those running chemical transformations involving singlet oxygen.

Because Reviewer 2 might have misunderstood some of the new data included in the last revision (particularly the method that employs the portion-wise addition of **17**), we have

modified the text in the revised manuscript to clarify our observations (changes highlighted in red):

‘Although a route for the preparation of artemisinin from dihydroartemisinic acid was reported as early as 1989,³² this transformation is challenging due to poor uniformity of the singlet oxygen production during the reaction time. Some elegant flow chemistry approaches have been reported to address this challenge, yet they require specialized equipment.³³ We envisioned that the controlled release of the PS **17** by AuPd resin-mediated uncaging of its protected analog **17b** would represent a simple alternative to continuously feed the reaction vessel with fresh compound **17** for the chemical synthesis of artemisinin in a one-pot manner. After optimization of the reaction conditions (Supplementary Note 2 in Supporting Information and Supplementary Figure 29), we found that irradiation (640 nm, 0.8 mW cm⁻²) of a water:methanol solution of dihydroartemisinic acid in the presence of compound **17b** (10% mol) and AuPd resins (2 mg mL⁻¹) afforded very high conversion rates to artemisinin (~80% after 10 h; Figures 5d and Supplementary Figure 30). Importantly, we also observed gradual uncaging of compound **17b** in the reaction mixture, supporting the advantageous effect of controlled singlet oxygen generation (Supplementary Figure 31). As controls, we performed analogous reactions under the same illumination regime using a single addition of equimolar amounts of uncaged PS **17** or compound **17b** in the absence of AuPd resins and found only traces of artemisinin in the reaction mixtures (Figure 5d). Finally, to mimic the continuous generation of PS **17** in the uncaging approach, the reaction was carried out under irradiation by adding the non-tunable phenothiazine **17** in a portion-wise manner (from 0.1 mM to 1 mM, 10% increment every hour). Under these conditions, the reaction yield improved to 69% (Supplementary Figure 32), in agreement with the hypothesis that singlet oxygen production is improved by adding or *in situ* generating fresh compound **17** during the course of the reaction. These results confirmed the importance of controlling singlet oxygen generation in some chemical transformations and indicate that our PS caging strategy can be employed to modulate complex photocatalyzed chemical reactions that require fine tuning of single oxygen production. The *in situ* gradual production of sensitive reagents within a closed reactor could be particularly helpful for procedures where the portion-wise addition of the reagent is difficult, such as reactions under inert atmosphere’.

Reviewer 3

The comments from this reviewer have been adequately addressed.

Answer: We are grateful to the Reviewer for pushing us to improve the work.

Reviewer 4

I appreciate the authors' additional work to demonstrate the stability of the probes and their explanations on the versatility of their system. I also agree that the scope of the strategy may be broader than it initially appears. However, I remain unconvinced that the manipulation of fluorescence response and photoinduced singlet oxygen generation of organic dyes through a cage-caging possesses the novelty and innovative elements required for publication in [journal name redacted] or Nature Communications. This is primarily because the underlying chemistry has been well established. While the work was competently performed and the results clearly presented, I believe that the manuscript would be better suited for a more specialized journal focusing on chemical biology or organic chemistry.

Answer: We thank the reviewer for the positive comments on the quality and rigor of our work as well as the clarity of the results, and also for acknowledging that the scope of the work is

broader upon revision. As stated by Reviewers 1, 2 and 3, we believe that publication in *Nature Communications* is fully justified.

We thank all the Reviewers for helping us to improve the quality of this manuscript.